# The Immunoproteasome Is Expressed but Dispensable for a Leukemia Infected Cell Vaccine

**DOI:** 10.3390/vaccines13080835

**Published:** 2025-08-05

**Authors:** Delphine Béland, Victor Mullins-Dansereau, Karen Geoffroy, Mélissa Viens, Kim Leclerc Desaulniers, Marie-Claude Bourgeois-Daigneault

**Affiliations:** 1Cancer Axis, Centre de Recherche du Centre Hospitalier de l’Université de Montréal and Institut du Cancer de Montréal, Montreal, QC H2X 0A9, Canada; delphine.beland@umontreal.ca (D.B.); victor.mullins-dansereau@umontreal.ca (V.M.-D.); ka.geoffroy@gmail.com (K.G.); melissa.viens@umontreal.ca (M.V.); kim.leclerc-desaulniers.chum@ssss.gouv.qc.ca (K.L.D.); 2Immunopathology Axis, Centre de Recherche du Centre Hospitalier de l’Université de Montréal, Montreal, QC H2X 0A9, Canada; 3Department of Microbiology, Infectious Diseases and Immunology, Faculty of Medicine, University of Montreal, Montreal, QC H3T 1J4, Canada

**Keywords:** immunoproteasome, oncolytic virotherapy, infected cell vaccine, vesicular stomatitis virus, leukemia

## Abstract

Background/Objectives: Leukemia is associated with high recurrence rates and cancer vaccines are emerging as a promising immunotherapy against the disease. Here, we investigate the mechanism of action by which a personalized vaccine made from leukemia cells infected with an oncolytic virus (ICV) induces anti-tumor immunity. Methods: Using the L1210 murine model, leukemia cells were infected and irradiated to create the ICV. The CRISPR-Cas9 system was used to engineer knockout cells to test in treatment efficacy studies. Results: We found that pro-inflammatory interferons (IFNs) that are produced by infected vaccine cells induce the immunoproteasome (ImP), a specialized proteasome subtype that is found in immune cells. Interestingly, we show that while a vaccine using the oncolytic vesicular stomatitis virus (oVSV) completely protects against tumor challenge, the wild-type (wt) virus, which does not induce the ImP, is not as effective. To delineate the contribution of the ImP for vaccine efficacy, we generated ImP-knockout cell lines and found no differences in treatment efficacy compared to wild-type cells. Furthermore, an ICV using another murine leukemia model that expresses the ImP only when infected by an IFN gamma-encoding variant of the virus demonstrated similar efficacy as the parental virus. Conclusions: Taken together, our data show that ImP expression by vaccine cells was not required for the efficacy of leukemia ICVs.

## 1. Introduction

Acute leukemias rank among the leading causes of death by cancer in children and young adults. The disease is aggressive, with 20–50% of the patients relapsing following initial remission in the United States [1,2]. Current leukemia treatment strategies such as chemotherapy and hematopoietic stem cell transplant are associated with severe toxicity for patients, coupled with a significant relapse rate [3,4]. There is, therefore, an urgent need for efficient strategies and immunotherapeutic approaches that are personalized to each patient in order to minimize the risk of autoimmunity. One promising strategy that confers complete protection in a murine model of the disease is the use of inactivated oncolytic virus (OV)-infected cancer cells that are administered as personalized vaccines [5].

OVs are viruses that preferentially replicate in cancer cells [6]. Their anti-tumor effects are the result of direct cancer killing, as well as the induction of systemic anti-tumor immunity [7]. For the vesicular stomatitis virus (VSV), a deletion of methionine 51 of its matrix protein (oVSV or VSVMΔ51) impairs its ability to counteract cellular antiviral defense mechanisms [6]. This enables oVSV to preferentially sustain replication in cancer cells, which commonly have defects in these pathways [8].

Infected cell vaccines (ICV) using oVSV or oncolytic maraba virus, another OV of the same family, have been shown to be effective against leukemia [5], melanoma [9], breast cancer [10] and peritoneal carcinomatosis [11]. The ICV is a personalized medicine approach for which each cancer is treated with a unique vaccine made from extracted tumor cells. The advantage of using an ICV rather than vaccinating against specific antigens is that it bypasses the laborious identification of cancer-specific immunogenic antigens and allows for immunization against all expressed antigens. Notably, as of today, ICV efficacy has only been demonstrated in murine models of cancer. While syngeneic cancer models are useful tools for pre-clinical studies testing novel therapies, they do not perfectly recapitulate human cancers, which are heterogeneous and for which environmental factors, patient diet and pre-treatment vary. Given these differences, ICV efficacy remains to be tested in cancer patients.

In murine models, several pre-clinical studies have shown that the efficacy of the approach requires CD8 T cells [5,10]. Nevertheless, little is known about the molecular mechanisms governing anti-tumor immunity following vaccination. Given that CD8 T cells, which recognize antigenic peptides that are presented at the cell surface by major histocompatibility complex class I (MHC-I) molecules, are required for ICV efficacy, we investigated the contribution of the antigen presentation machinery for treatment efficacy.

The first step of antigen presentation is the proteasomal processing of proteins into peptides. While the constitutive proteasome, which is expressed by most cells, has chymotrypsin-, caspase- and trypsin-like cleavage activities, the immunoproteasome (ImP), which is typically expressed by immune cells but can also be induced by other cells under inflammatory conditions, only performs chymotrypsin- and trypsin-like cleavages [12]. As such, the ImP executes twice as much chymotrypsin, but no caspase-like cleavage [12]. Importantly, chymotrypsin-like cleavage yields hydrophobic C-termini, which are optimal for binding to the MHC-I peptide-binding niche. The type of proteasomes that are found in a cell therefore affects the surface repertoire of peptides [13]. The enzymatic activities of the different proteasomes are determined by their compositions, and they contain three catalytic subunits: PSMB5, 6 and 7 for the constitutive proteasome and PSMB8, 9 and 10 for the ImP. Interestingly, previous studies have shown the importance of ImP epitopes for leukemia-immunogenicity. Indeed, the immuno-dominant epitope of Wilm’s tumor antigen 1, an antigen that is often found in leukemia, was downregulated in a patient with low ImP expression and PSMB9-expressing cells were selectively lost after adoptive T cell therapy against the antigen [14]. Furthermore, the BCR-ABL oncogene is an antigenic fusion protein found in leukemia for which it was shown that ImP-mediated protein degradation generated a higher amount MHC-I-compatible peptides [15]. Given that oVSV triggers inflammation, which is known to induce ImP expression and considering the established importance of the ImP for MHC-I antigen presentation, we investigated its contribution to ICV efficacy.

Here, we demonstrate that leukemia cells infected with oVSV produce type-I interferons (IFNs) upon infection, which in turn induce the ImP. In cell lines that do not produce IFNs, an oVSV variant that was engineered to express IFN gamma (IFNγ) could trigger ImP expression. We also show that infection by wild-type (wt)VSV does not induce IFN production nor ImP expression. Finally, blocking IFN in the context of oVSV infection effectively prevented ImP induction. When testing the in vivo efficacy of the treatment, we found the ICV using wtVSV to be less protective compared to oVSV, an effect that was independent from the ImP as an ImP-KO vaccine conferred similar therapeutic efficacy compared to control (ctrl) cells. Taken together, our data show that the ImP is induced by IFN-competent tumor cells upon infection but not required for treatment efficacy.

## 2. Materials and Methods

### 2.1. Cell Culture

L1210 murine lymphoblastic leukemia, African green monkey Vero cells (both from the American Type Culture Collection) and C1498 murine myeloid leukemia cells (kindly provided by Dr Michele Ardolino, Ottawa Hospital Research Institute) were cultured in Dulbecco’s modified Eagle medium (DMEM; Gibco, Billings, MT, USA) supplemented with 10% fetal bovine serum (FBS; Wisent, St-Bruno, QC, Canada). EL4 murine T lymphoma cells (kindly provided by Dr Claude Perreault, Institute for Research in Immunology and Cancer) were cultured in RPMI-1640 medium supplemented with 10% FBS (both from Wisent), 2 mM L-glutamine, 1 mM HEPES, 1 mM sodium pyruvate, 100 units/mL penicillin-streptomycin (all from Gibco) and 1% minimum essential medium non-essential amino acids (Wisent). All cells were maintained at 37 °C with 5% CO_2_.

### 2.2. IFN Treatment of Leukemia Cells

For ImP induction experiments, cells were stimulated for 24 h with recombinant murine IFNα, IFNβ (both from PBL Assay Science, Piscataway, NJ, USA) or IFNγ (Peprotech, Cranbury, NJ, USA) at concentrations of 1000, 200 and 1000 units/mL, respectively, as performed in previous studies by other groups [16,17]. For virus protection experiments, cells were stimulated with the different IFNs for 4 h prior to infection.

### 2.3. Viruses and Virus Assays

The wtVSV [8] and oVSV mutant (VSVΔ51) [6] used in this study are Indiana strain variants that were genetically engineered to encode either the green or yellow fluorescent protein (YFP or GFP), respectively, or IFNγ [18]. Virus stocks were propagated in Vero cells as described elsewhere [19].

Monitoring of infection was performed by fluorescence imaging using a ZOE™ fluorescent cell imager (Biorad, Hercules, CA, USA).

Viral titers were measured by plaque assay. Briefly, serial dilutions of infectious samples were prepared in serum-free DMEM and used to infect monolayers of Vero cells for 1 h, prior to overlaying a solution of 1% agarose (Thermofisher, Waltham, MA, USA) in DMEM supplemented with 10% FBS. Plaques were enumerated 24 h later.

### 2.4. Western Blot

For experiments using CM, supernatants from leukemia cells infected for 24 h at MOIs of 10 (L1210) or 1 (Cl498 and EL4), were filtered using Amicon^®^ Ultra-15 centrifugal filter units with 100kDa molecular weight cutoffs (Millipore Sigma, Burlington, MA, USA) to remove viral particles. For type I IFN blockade experiments, cells were co-treated with virus CM and an anti-IFNAR-1 antibody (BioXCell clone MAR1-5A3, Lebanon, NH, USA) as described in the figure legends.

Cell lysates were generated using Mammalian Protein Extraction Reagent (MPER; Thermofisher Scientific, Waltham, MA, USA) supplemented with EDTA-free complete protease inhibitors (Roche, Basel, Switzerland) according to the manufacturers’ protocols. Cleared lysates were then mixed with Laemmli loading buffer (312.5 mM Tris-HCl (Wisent), 5% β-mercaptoethanol (Gibco), 0.0125% bromophenol blue (Fisher Chemical, Pittsburgh, PA, USA), 2% sodium dodecyl sulfate (Bioshop, Burlington, ON, Canada) and 50% glycerol (Fisher Scientific, Waltham, MA, USA), ran on 12% polyacrylamide gels (Thermofisher) in Tris/Glycine/SDS buffer (Biorad) and transferred onto 0.45 µm nitrocellulose membranes (Biorad) in Tris/Glycine buffer (Biorad). Membranes were then blocked in 5% non-fat dry milk (No Name brand, Brampton, ON, Canada) diluted in Tris buffered saline (20 mM Tris base, 150 mM NaCl, pH = 7.6) supplemented with 0.1% Tween20 (Sigma Aldrich, St-Louis, MO, USA) (TBS-T-milk) and incubated overnight with rabbit anti-mouse PSMB8 (ab3329, diluted 1/500), 9 (ab3328, diluted 1/1000), 10 (ab183506, diluted 1/1000), PSMA4 (ab191403, diluted 1/2000) (all from Abcam, Cambridge, UK), GAPDH (2118S, diluted 1/1000), β-actin (4967S, diluted 1/1000) (both from Cell Signaling Technologies, Danvers, MA, USA) or anti-VSV (made in-house, diluted 1/10 000 [18]) primary antibodies. A goat-anti-rabbit-HRP IgG (7074S, diluted 1/2000, Cell Signaling Technologies, Danvers, MA, USA) was used as a secondary antibody. All antibodies were diluted in TBS-T-milk. Chemiluminescent signals were acquired using Clarity Max™ Western ECL Substrate (Biorad) and a ChemiDoc imager (Biorad).

For signal quantification, band intensities were determined using the ImageJ software (version 1.54p). Briefly, each band was defined as a region of interest, the corresponding integrated density values were obtained, and membrane backgrounds were subtracted. Ratios of PSMB8, 9 and 10 signals over PSMA4 expression were calculated to normalize the expression of ImP subunits to proteasome content.

### 2.5. Proteasome Isolation

Proteasomes were isolated using a Proteasome Isolation Kit (Millipore Sigma, Burliington, MA, USA) as per the manufacturer’s instructions. Briefly, pull-downs using the mature ImP regulatory cap subunit PSMD4 were performed on cell lysates for 4 h at 4 °C and eluted in Laemmli loading buffer prior to Western blotting.

### 2.6. ICV Preparation

The vaccines were prepared by infecting tumor cells at an MOI of 1 (EL4 and C1498) or 10 (L1210) for 18 h at 37 °C with 5% CO_2_. Infected cells were then centrifugated at 500× *g* for 5 min at 4 °C, the pellets were washed once using phosphate-buffered saline (PBS; Gibco) and resuspended in PBS at a final concentration of 10^7^ cells/mL. The cells were then γ-irradiated using a Gammacell 1000 Elite/3000 Elan (Best Theratronics, Kanata, ON, Canada) at 30 Gy a dose, at which L1210 cells were shown to lose their replicative ability while preserving their cytokine-production capacity [20]. L1210 and EL4 ICVs prepared following the same methodology were shown to be protective and not allow for tumor cell proliferation [5].

### 2.7. In Vivo Experiments

DBA/2 female mice were purchased from Jackson Laboratories and C57BL/6 mice from Charles River Laboratories (all 6–8 weeks old). Animals were housed in the CRCHUM’s animal facility, and all in vivo studies were performed in accordance with institutional guidelines. All efficacy studies included at least 10 mice/group, as described in the figure legends. Given that some animals were administered infectious material, blinding was not performed for safety reasons. Over the course of the experiments, 2 mice experienced severe malocclusions, which prevented proper feeding and resulted in body weight loss. These animals were excluded from our analysis. No other exclusion criteria were applied.

For prophylactic experiments, mice were vaccinated at days −21, −14 and −7 with 100 µL of ICV or PBS intravenously (IV) in the tail vein and challenged on day 0 with 10^6^ cells of the corresponding cell line. For therapeutic experiments, mice were challenged with 10^6^ cells on day 0, and vaccinated on days 1, 8 and 15, or on days 3, 10 and 17, as depicted in the timelines and described in the corresponding figure legends. Mice were monitored for advanced symptoms of leukemia such as lethargy, the apparition of palpable nodules, as well as hind limb paralysis, at which point they were euthanized.

Tumor models, mouse strains, experimental details and outcomes of each experiment are summarized in Appendix A.

### 2.8. CRISPR-Cas9 Genome Editing

ImP subunit genes *Psmb8*, *9*, and *10* were KO using the pSpCas9 (BB)-2A-GFP vector system from Addgene. Oligonucleotide pairs encoding for short guide RNAs were cloned into the vector backbone as previously described in Ran; et al. [21]. Sequences for PSMB8: 5′-CACCGCTCGCCTTCAAGTTCCAGCA-3′ and 5′-AAACTGCTGGAACTTGAAGGCGAGC-3′; for PSMB9: 5′-CACCGAGGTATATGGAACCATGGGA-3′ and 5′-AAACTCCCATGGTTCCATATACCTC-3′; and for PSMB10: 5′-CACCGCACTAACGATTCGGTTGTGG-3′ and 5′-AAACCCCACAACCGAATCGTTAGTCC-3′.

Cells were transfected with the different CRISPR plasmids using lipofectamine 2000 (Thermofisher, Waltham, MA, USA) and GFP+ cells were sorted using a BD FACSAria™ cell sorter 24 h later. Individual clones were expanded, and KOs were validated by Western blot.

### 2.9. Enzyme-Linked Immunosorbent Assays (ELISAs)

ELISAs for IFNα were performed using a commercial kit (PBL Assay Science, Piscataway, NJ, USA) following the manufacturer’s instructions. For IFNβ ELISAs, a capture antibody (Biolegend clone Poly5192, San Diego, CA, USA) diluted in PBS and a biotin-conjugated detection antibody (Biolegend, MEB-5E9.1, San Diego, CA, USA) diluted in PBS containing 1% bovine serum albumin (BSA) were used, as well as HRP-coupled streptavidin (R&D Systems, Minneapolis, MN, USA) diluted in PBS-1% BSA. A commercial IFNβ standard (Biolegend catalog 581309, Recombinant Mouse IFN-β (ELISA standard) San Diego, CA, USA) [22], TMB substrate (Millipore Sigma, Burliington, MA, USA) and stop solution (R&D Systems) were used according to manufacturer’s instructions. Assays were performed in 96-well EIA/RIA high-binding plates (Corning, Corning, NY, USA), and optical density results were obtained using an Ensight™ multimode plate reader (Perkin Elmer, Waltham, MA, USA) set at 450 nm. Readings obtained at 570 nm were subtracted to correct for optical imperfections.

### 2.10. Statistical Analyses

All analyses were performed as described in the figure legends using the Prism 9 software.

## 3. Results

### 3.1. The ImP Is Induced by oVSV Infection in L1210 Cells

For the ImP to be important for the efficacy of the ICV, it must be expressed by vaccine cells. Our hypothesis is that pro-inflammatory factors are produced upon infection and in turn induce ImP expression (Figure 1A).

To validate our model, we first confirmed that L1210 cells were permissive to oVSV infection. Using a YFP-engineered variant of oVSV, we could detect fluorescence 24 h post-infection (Appendix A) and infectious virus particles were produced over time by L1210 cells infected at low (0.01) or high (10) MOIs (Appendix A, respectively). We then assessed IFN production and found that IFNα and β were both produced by infected L1210 cells (Figure 1B,C). We then measured IFN responsiveness by measuring the anti-viral protective capacity of IFNs in L1210 cells. To do so, cells were pre-treated with IFNα, β or ɣ for 4 h and then infected and virus production was measured by plaque assay. Our results show that all 3 IFNs could lower virus production, therefore indicating that the cells are IFN responsive (Figure 1D). Finally, we confirmed that IFNs could induce ImP expression in L1210 cells and found the all three ImP subunits (PSMB8, 9 and 10) were induced by IFNα, β and ɣ stimulations (Figure 1E). PSMA4, a subunit that is found in all proteasome subtypes, was used as a control for the amount of proteasomes in our lysates, which is stable in all conditions. Taken together, our data support our model.

Next, we tested whether oVSV infection could induce ImP expression. In line with our model, we found that oVSV infection induced the expression of PSMB8, 9 and 10 (Figure 1F and Appendix A). Importantly, in order to form functional ImPs, the subunits must be incorporated into the proteasome complex. To assess incorporation, we isolated proteasomes from cell lysates and performed a Western blot to detect the subunits. Our data show that PSMB8, 9 and 10 are found in both cell lysates and purified proteasomes upon virus infection, therefore indicating that they are incorporated into the complex (Figure 1G and Appendix A). Altogether, we found that the ImP is induced and assembled in L1210 cells upon oVSV infection.

### 3.2. ImP Induction by Infected L1210 Cells Is Type I IFN-Mediated

We next wanted to determine if soluble factors produced by L1210 cells in response to oVSV infection were mediating ImP induction. To do so, we used conditioned media (CM) from virus-infected cells and removed the virus by filtration. As such, the CM contains the soluble factors that were produced by the cells, but no infectious viral particles. Our results show comparable induction of PSMB8, 9 and 10 upon stimulation with CM and direct infection (Figure 2A). Since wtVSV, the parental strain of oVSV, has been reported to prevent the production of type I IFNs [23], we next tested its ability to induce the ImP. First, we confirmed that, as opposed to what we observed with oVSV, IFNα and IFNβ were not produced post-wtVSV infection (Appendix A). In line with IFNs being required for ImP induction, PSMB8 and 9 were not induced by wtVSV infection, and while the signal for PSMB10 was slightly increased by wtVSV, the expression remained lower compared to oVSV (Figure 2B). Altogether, our data show that soluble factors produced by tumor cells upon oVSV infection induce ImP expression.

Given that IFNs are established inducers of the ImP [24], we next wanted to determine if they were mediating this effect upon infection. To do so, we stimulated cells with CM from oVSV-infected cells and co-treated with an IFNAR-1-blocking antibody, which prevents stimulation by both IFNα and β. We found that blocking type I IFNs completely abrogated the induction of PSMB8 and 9 (Figure 2C). PSMB10 expression was also found to be lower in this condition. These results are in line with our working model and confirm that type I IFNs are produced upon oVSV infection and induce ImP expression in L1210 cells.

### 3.3. wtVSV-ICV Is Less Protective Compared to oVSV-ICV

Before testing the efficacy of various ICV preparations, we first validated that ɣ-irradiation, which prevents vaccine cells from proliferating and is required for ICV preparation, did not hamper ImP expression. To do so, the ICV was prepared by infecting L1210 cells for 18 h prior to ɣ-irradiation and then allowing for virus production for another 24 h. Our results show that infected irradiated cells express increased levels of PSMB8, 9 and 10 both upon irradiation and 24 h later (Appendix A). We also found that irradiated cells do not produce infectious viral particles post-irradiation (Appendix A). Given that the ImP is expressed by oVSV- but not wtVSV-infected L1210 cells, we postulated that if the ImP was important for the efficacy of the vaccine, an ICV prepared with wtVSV should be less effective. To compare the therapeutic efficacies of wtVSV- and oVSV-ICVs, we used a prophylactic immunization model, which has been previously shown to be almost 100% protective by another group [5]. As depicted in Figure 3A, mice were vaccinated 7, 14 and 21 days before tumor challenge and Kaplan–Meier survival analyses were conducted. As expected, we found that 100% of oVSV-ICV-vaccinated mice were protected against tumor challenge (Figure 3B). Also, in line with our hypothesis, the survival was decreased for the group that received the wtVSV-ICV vaccine.

To extend our findings to additional murine models, we next tested the EL4 T lymphoma and C1498 acute myeloid leukemia cell lines. We first validated that both cell lines were permissive to viral infection (Appendix A) and next tested their ability to produce and respond to IFN. While we found that stimulation with exogenous IFNs could induce PSMB8, PSMB9 and PSMB10 expression in both cell lines (Appendix A), neither could produce the cytokines upon virus infection (Appendix A). As expected, given that infection does not lead to IFN production in these cell lines, oVSV infection and CM from infected cells did not induce the ImP (Appendix A). Notably, while the virus failed at inducing the ImP in EL4 cells, the basal levels of PSMB8, 9 and 10 expression were high at baseline. These results are summarized in Table 1. Interestingly, a previous study has reported that the M protein of wtVSV prevents PSMB9 incorporation into the proteasome and therefore interferes with ImP function [25]. We therefore postulated that if ImP function was required for ICV efficacy, an EL4 ICV, which expresses high ImP at baseline, it would be less protective using wtVSV compared to oVSV, which we show in Figure 1C does not prevent the incorporation of PSMB9 into the proteasome. To test this, we prophylactically immunized C57BL/6 mice with EL4 ICVs prepared with either virus (Figure 3A) and observed a non-statistically significant trend towards a decrease in survival for wtVSV-ICV-vaccinated mice compared to the oVSV-ICV group (Figure 3C). Overall, our data using both the L1210 and EL4 models show that wtVSV is not as effective as oVSV in the ICV setting, which could result from the lack of ImP expression after infection or other differences in the cellular responses to both viruses.

### 3.4. oVSV-IFNɣ Enhances ImP Expression but Does Not Increase ICV Efficacy

While prophylactic ICV vaccination confers 100% protection against L1210 leukemia, therapeutic vaccination is usually less effective [5]. We therefore tested if further enhancing ImP expression could improve vaccination efficacy in this setting. To do so, we exploited an oVSV variant that was engineered to express IFNɣ (oVSV-IFNɣ) [18], a master inducer of the ImP [26]. As expected, L1210 cells infected with oVSV-IFNɣ expressed higher levels of PSMB8, 9 and 10 compared to cells infected with the parental virus (Figure 4A and Appendix A). We then compared oVSV- and oVSV-IFNɣ-ICVs in a therapeutic setting in which mice were vaccinated on days 1, 8 and 15 post-challenge (Figure 4B). Our results show that both ICVs were equally effective (Figure 4C). We next repeated the experiment in a more aggressive setting, in which the tumors were allowed to establish for two more days prior to initiating ICV treatments and once again found that both vaccines conferred similar therapeutic efficacies (Appendix A). Taken together, our data show no therapeutic advantage to further enhancing ImP expression in conditions where the ImP is already induced.

Given that IFN stimulation, but not oVSV infection of C1498 cells that do not produce IFNs, could induce ImP expression, we next tested whether oVSV-IFNɣ could improve ICV therapeutic efficacy in this model. As expected, C1498 cells treated with oVSV-IFNɣ CM exhibited striking ImP induction compared to oVSV CM (Figure 4D and Appendix A). We then tested the vaccines in a prophylactic setting and found that both oVSV and oVSV-IFNɣ conferred modest therapeutic efficacies in this context (Figure 4E), therefore demonstrating that IFNɣ expression or ImP expression did not benefit the C1498 ICV.

### 3.5. The ImP Is Not Required for ICV Efficacy

Given that our results support a potential role of the ImP in the L1210 ICV but not in other models, we next directly tested its role by using cells that cannot induce ImP expression. To do so, we used the CRISPR-Cas9 system to engineer L1210 cells lacking PSMB8, 9 and 10 (L1210ImP-KO). Unlike parental cells (L1210ctrl), L1210ImP-KO cells should not induce ImP subunits following oVSV infection and will therefore only contain constitutive proteasomes (Figure 5A).

We first validated that our L1210ImP-KO cells could not induce PSMB8, 9 and 10 following IFNγ stimulation (Figure 5B) and that their proliferation (Appendix A), oVSV production upon infection (Appendix A), IFNα and IFNβ production capacities (Appendix A) and responses (Appendix A) were similar to L1210ctrl cells. Next, we tested if the lack of ImP expression by vaccine cells could impact treatment efficacy by prophylactically vaccinating mice with either L1210ctrl- or ImP-KO-ICV and then challenging with L1210ctrl cells. While 100% of the mice vaccinated with L1210ctrl-ICV were protected against tumor challenge, 20% of the L1210ImP-KO-ICV group did not survive, a difference that was not statistically significant (Figure 5C). Since the prophylactic vaccination model is highly protective, which could mask a slight decrease in treatment efficacy, we also tested the vaccine in a more aggressive therapeutic setting. As was observed for prophylactic vaccination, we did not observe significant differences in mouse survival following L1210ImP-KO- vs. L1210ctrl-ICV treatments (Figure 5D). We then tested if ImP expression by tumor cells rather than vaccine cells was required for protection by prophylactically vaccinating mice with the oVSV-ICV and then challenging with either L1210ctrl or ImP-KO cells. Interestingly, the vaccine conferred 100% protection against both ctrl and ImP-KO tumor cells (Figure 5E). We rechallenged the mice again at day 100, and while we lost 2 mice per group with this second challenge, we once again failed to observe differences in mortality rates, therefore indicating that ImP expression by tumor cells is not required for protection. Collectively, our data show that the ImP is neither required for vaccination efficacy nor tumor clearance.

## 4. Discussion

Over the last decade, the ImP has emerged as a promising target in cancer. Its central role in MHC-I antigen presentation makes it a potential player for ICV efficacy. Indeed, previous reports show that increased ImP expression favors the presentation of tumor antigens in a variety of models [27,28,29]. Given that leukemias rank amongst the least mutated cancers [30], optimal processing of the limited tumor antigen pool could be particularly important. Here, we postulated that ImP expression by vaccine cells contributes to ICV efficacy. While we show ImP induction in IFN competent leukemia cells following oVSV infection, ICVs prepared with ImP-KO cells were as effective as parental cells, therefore demonstrating that the ImP is not required in our model. Notably, leukemia cells are hematopoietic stem cells and their proteasome content is mainly composed of ImPs. Given that ImP KO did not alter ICV efficacy, our results suggest that the antigenic peptides derived from the constitutive proteasome are sufficient to induce anti-tumor immunity. Although our results did not reveal a role for the ImP in anti-tumor immunity following leukemia ICV vaccination, it might be important in the context of cancers with a higher mutational burden.

In this study, we show that unlike wtVSV, oVSV induces ImP expression via type I IFN production and that wtVSV-ICVs are less effective compared to oVSV-ICVs. While we initially hypothesized that the decreased efficacy of the wtVSV-ICV could result from the lack of ImP expression or assembly, our results obtained with the ImP-KO-ICV indicate the opposite. Therefore, other factors that differ between oVSV- and wtVSV-ICVs must affect efficacy. While the M protein of wtVSV blocks the nucleocytoplasmic transport of host mRNAs, and therefore drastically impairs the transcription of host genes [31,32], it also dampens the master pro-inflammatory transcription factor nuclear factor kappa-light-chain-enhancer of activated B cells (NF-κB) [33] and signal transducer and activator of transcription (STAT) activation [34], which induces the expression of multiple cytokines. As such, the decreased efficacy observed using wtVSV could be the result of lower inflammation following treatment.

We also observed that, while oVSV-IFNɣ enhanced ImP expression by L1210 and C1498 cells, it did not improve treatment efficacy compared to oVSV. Notably, the C1498 model is particularly aggressive with all mice reaching endpoint within 60 days post-challenge even with prophylactic vaccination. This limited protection suggests poor anti-tumor immunity or cancer cell recognition. Interestingly, C1498 cells have been previously reported to display high clonal heterogeneity resulting in decreased immune surveillance by CD8 T cells and a high potential for immune escape [35,36,37]. Also, C1498 cells are poorly immunogenic, a feature that is often found in leukemia. Indeed, leukemia ranks amongst the cancer types that have the lowest mutational burdens [30], a context in which the ICV is not likely to reach its full potential.

Overall, we show that ImP expression by vaccine or challenge cells is not required for treatment efficacy. Future studies will aim at further exploring how OV infection links the ICV to the anti-tumor immune response, as well as measuring its importance in tumor models with higher mutagenic burdens compared to leukemia. For instance, melanoma has a high mutational burden and the ICV was shown to be effective against the B16F10 murine melanoma cell line [9]. Similarly, colon cancer, which is also more mutated compared to leukemia, can also be effectively treated using an ICV approach in mice [11]. Future studies will investigate the importance of the ImP for ICV efficacy in these tumor models.

## Figures and Tables

**Figure 1 vaccines-13-00835-f001:**
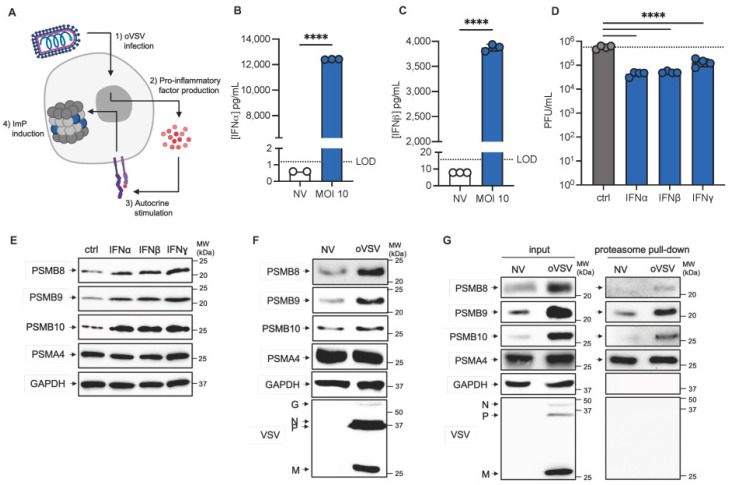
The ImP is induced by oVSV infection. (**A**) Schematic representation of the working model of our study. The virus triggers the production of pro-inflammatory factors, which induce ImP expression in an autocrine manner. (**B**) IFNα and (**C**) IFNβ production by L1210 cells 24 h post-infection with oVSV-YFP at an MOI of 10 as measured by ELISA (*n* = 2–3 for NV and 3 for oVSV-YFP). Data are representative of two independent experiments. (**D**) Virus production by L1210 cells treated with IFNα, IFNβ or IFNɣ for 4 h and then infected for 24 h with oVSV-YFP at an MOI of 0.1 (*n* = 4). The dotted line indicates the average virus production for the control condition (representative of two independent experiments). **** *p* < 0.001 (one-tailed unpaired *t*-test with Welch’s correction). Non-statistically significant differences (*p* ≥ 0.05) are not indicated on the graphs. Western blot analysis of PSMB8, 9 and 10 expression by (**E**) L1210 cells stimulated for 24 h with IFNα, IFNβ or IFNɣ (representative of two independent experiments), (**F**) L1210 cells infected for 24 h with oVSV-YFP at an MOI of 10 (representative of 4 independent experiments) and (**G**) L1210 lysates and isolated proteasome from samples treated as in B (representative of two independent experiments).

**Figure 2 vaccines-13-00835-f002:**
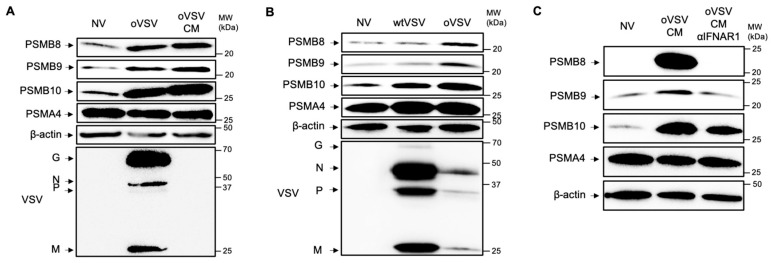
Type I IFNs mediate ImP induction upon oVSV infection. Western blot analysis of PSMB8, 9 and 10 expression of L1210 cells (**A**) infected for 24 h with oVSV-YFP at an MOI of 10 or stimulated for 24 h with virus-cleared CM from cells infected in the same conditions (representative of three independent experiments), (**B**) infected for 24 h with oVSV-YFP or wtVSV-GFP at an MOI of 10 or (**C**) stimulated with virus-cleared CM as in A, with or without IFNAR1 blocking antibody (αIFNAR1). Data are representative of three independent experiments.

**Figure 3 vaccines-13-00835-f003:**
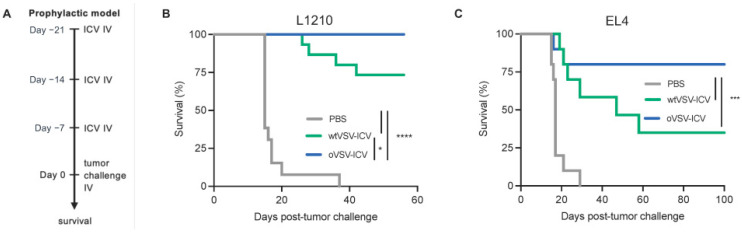
oVSV-ICV is more protective compared with wtVSV-ICV. (**A**) Experimental timeline of the prophylactic ICV vaccination regimen used in this study. Kaplan–Meier survival analyses of mice prophylactically vaccinated with (**B**) PBS (mock) (*n* = 13), wtVSV-GFP-L1210-ICV (*n* = 15) or oVSV-YFP-L1210-ICV (*n* = 14) and then challenged with L1210 cells (representative of two independent experiments) or (**C**) PBS (mock) (*n* = 10), wtVSV-GFP-EL4-ICV (*n* = 10) or oVSV-YFP-EL4-ICV (*n* = 10) and then challenged with EL4 cells. The experiment was performed once. * *p* < 0.05, *** *p* < 0.001, **** *p* < 0.0001 (log rank Mantel–Cox test). Non-statistically significant differences (*p* ≥ 0.05) are not indicated on the graphs.

**Figure 4 vaccines-13-00835-f004:**
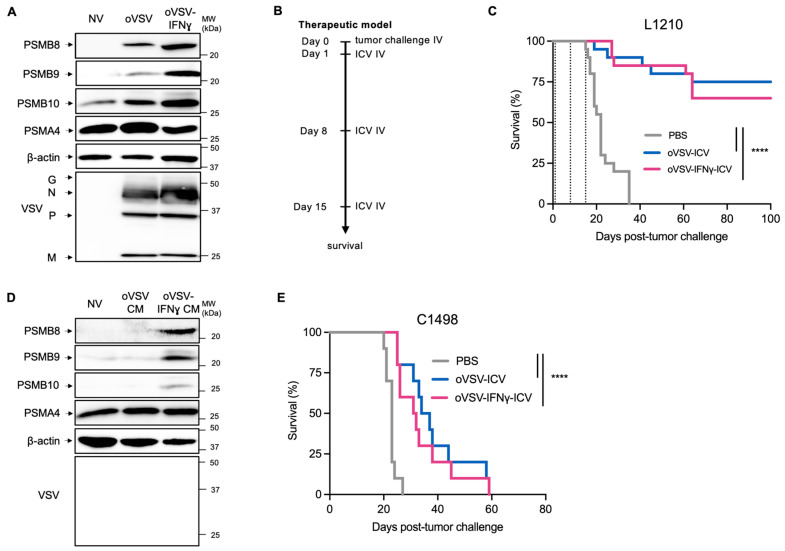
Enhanced ImP expression by oVSV-IFNɣ does not improve ICV efficacy. (**A**) Western blot analysis of PSMB8, 9 and 10 expression by L1210 cells 18 h post-infection with oVSV-YFP or oVSV-IFNɣ at an MOI of 10 (representative of two independent experiments). (**B**) Experimental timeline of the therapeutic ICV vaccination regimen used in this study. (**C**) Kaplan–Meier survival analysis of L1210-challenged mice treated with PBS (mock, *n* = 20), oVSV-YFP-ICV (*n* = 20) or oVSV-IFNɣ-ICV (*n* = 19). The dotted lines indicate the days of treatment (representative of two independent experiments). (**D**) Western blot analysis of PSMB8, 9 and 10 expression of C1498 cells 24 h post-treatment with CM from oVSV-YFP or -IFNɣ-infected C1498 cells. Data are representative of two independent experiments. (**E**) Kaplan–Meier survival analysis of mice prophylactically vaccinated with PBS (mock, *n* = 10), oVSV-YFP-ICV (*n* = 10) or oVSV-IFNɣ-ICV (*n* = 10) and then challenged with C1498 cells. The experiment was performed once. **** *p* < 0.0001 (log rank Mantel–Cox test). Non-statistically significant differences (*p* ≥ 0.05) are not indicated on the graphs.

**Figure 5 vaccines-13-00835-f005:**
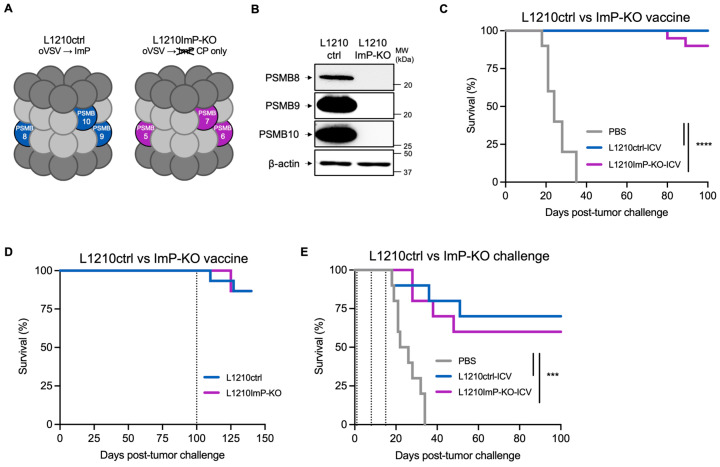
The ImP is not required for ICV efficacy. (**A**) Schematic representation of proteasomes from L1210ctrl and ImP-KO cells after oVSV-YFP infection. (**B**) Western blot validation of PSMB8, 9 and 10KO of L1210ImP-KO cells stimulated with IFNɣ for 24 h (representative of two independent experiments). Kaplan–Meier survival analyses of mice (**C**) prophylactically vaccinated with PBS (mock, *n* = 20), L1210ctrl-ICV (*n* = 20) or L1210ImP-KO-ICV (*n* = 20), and then challenged with L1210ctrl cells (representative of two independent experiments), (**D**) challenged with L1210ctrl cells and then vaccinated with PBS (mock, *n* = 10), L1210ctrl-ICV (*n* = 10) or L1210ImP-KO-ICV (*n* = 10) (the experiment was performed once) or (**E**) prophylactically vaccinated with L1210ctrl-ICV and then challenged on day 0 with L1210ctrl (*n* = 15) or L1210ImP-KO cells (*n* = 15) and re-challenged again on day 100. The experiment was performed once. Dotted lines represent the days of treatment (panel D) or the days of the re-challenge (panel E). *** *p* < 0.0001 and **** *p* < 0.0001 (log rank Mantel–Cox test). Non-statistically significant differences (*p* ≥ 0.05) are not indicated on the graphs.

**Table 1 vaccines-13-00835-t001:** ImP induction by IFNs in hematological cancer cell lines.

		L1210	EL4	C1498
IFN production	IFNα	+	−	−
	IFNβ	+	−	−
	IFNɣ	−	−	−
Baseline expression all 3 subunits		+	+	−
PSMB8 induction	IFNα	−	+	+
	IFNβ	+	+	+
	IFNɣ	+	−	+
	oVSV	+	−	−
PSMB9 induction	IFNα	−	+	+
	IFNβ	+	+	+
	IFNɣ	+	−	+
	oVSV	+	−	−
PSMB10 induction	IFNα	+	+	+
	IFNβ	+	+	+
	IFNɣ	+	+	+
	oVSV	+	−	−

IFN production was measured by ELISA and expression/induction of PSMB8, 9 and 10 were measured by Western blot 24 h after stimulation with the different IFNs. +: the subunit is expressed at higher levels post-stimulation. −: the stimulation did not modulate expression.

## Data Availability

Data is contained within the article or Appendix A. Further inquiries can be directed to the corresponding author.

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
