# Peer review of "The Immunoproteasome Is Expressed but Dispensable for a Leukemia Infected Cell Vaccine"

_vaccines, 2025, doi:10.3390/vaccines13080835_

Round 1

Reviewer 1 Report

Comments and Suggestions for Authors

Review of "The immunoproteasome is expressed but dispensable for a leukemia infected cell vaccine"

General Comments:

This manuscript presents a comprehensive investigation into the role of the immunoproteasome (ImP) in the efficacy of oncolytic vesicular stomatitis virus (oVSV)-infected cell vaccines (ICVs) against leukemia. The authors rigorously demonstrate that while ImP expression is induced in IFN-competent leukemia cells following oVSV infection, its presence is ultimately dispensable for the therapeutic efficacy of ICVs in this context. The manuscript is well-organized and clearly written, and the experimental approach is thorough and thoughtfully executed.

Major Points:

  1. Introduction and Rationale:
    The introduction successfully outlines the rationale for exploring the role of the ImP; however, it would benefit from expanded discussion on why current leukemia treatments (e.g., stem cell transplantation, targeted therapies) are insufficient and what unmet clinical needs this vaccine strategy aims to address. Further, while the authors correctly state that 20–50% of patients relapse, they should clarify whether this statistic applies globally or is specific to a certain region (e.g., the United States).
  2. Mechanistic Insights and Immune Correlates:
    While the study effectively demonstrates that ImP is not required for ICV efficacy, the manuscript lacks direct assessment of adaptive immune responses, particularly T cell activation or infiltration. Profiling CD8+ T cells, evaluating tumor-infiltrating lymphocytes, or measuring antigen-specific responses would strengthen the mechanistic link to MHC-I presentation and bolster conclusions on how these vaccines drive anti-tumor immunity.
  3. Clarification of wtVSV-ICV Efficacy:
    The decreased efficacy of wtVSV-ICV is noted, with speculation on lower inflammation or impaired antigen presentation due to the M protein. However, the manuscript does not directly test these hypotheses. Future experiments investigating cytokine profiles, inflammatory gene signatures, or T cell responses in this context would help clarify this mechanistic gap.
  4. Figure Analysis and Quantification:
    Some western blots (e.g., Figures 2B and 2C) are interpreted as indicating partial induction of PSMB10, yet these claims would be more robust if supported by densitometric quantification normalized to loading controls and presented with statistical analysis. Persistent PSMB10 expression despite IFNAR1 blockade suggests alternative regulatory mechanisms, meriting further investigation (e.g., mRNA analysis, protein stability assays).
  5. Formatting and Presentation:
    The overall figure quality is low in print; sharper diagrams with clearer labeling are recommended to improve readability. Additionally, minor typographical corrections (e.g., "stain" vs. "strain," "O" in "complete protease inhibitors") should be addressed.

Minor Points:

  • There is a formatting issue in section 3.2 (visible gap in paragraph start), which should be corrected for clarity.
  • Line 291–293 suggests a "potential role for ImP expression" in efficacy, which appears inconsistent with the conclusion that ImP is dispensable. This statement should be revised or clarified.
  • The authors might wish to explicitly discuss the limited generalizability of these findings to human leukemia or to cancers with higher mutational burdens, as acknowledged in the discussion.

Conclusions:

This study contributes valuable negative data to the field, challenging assumptions regarding the necessity of the immunoproteasome in infected cell vaccine efficacy. It reinforces that while ImP expression is induced and theoretically advantageous for antigen processing, it is not essential in the leukemia models tested. Further mechanistic work examining adaptive immune correlates and extending to more heterogeneous or highly mutated cancers would provide critical insights and improve translatability to clinical settings.

Comments on the Quality of English Language

adequate

Author Response

General comment:

  1. This manuscript presents a comprehensive investigation into the role of the immunoproteasome (ImP) in the efficacy of oncolytic vesicular stomatitis virus (oVSV)-infected cell vaccines (ICVs) against leukemia. The authors rigorously demonstrate that while ImP expression is induced in IFN-competent leukemia cells following oVSV infection, its presence is ultimately dispensable for the therapeutic efficacy of ICVs in this context. The manuscript is well-organized and clearly written, and the experimental approach is thorough and thoughtfully executed.”

We are glad that the reviewer appreciated the quality of our work.

  1. Major points: Introduction and Rationale: The introduction successfully outlines the rationale for exploring the role of the ImP; however, it would benefit from expanded discussion on why current leukemia treatments (e.g., stem cell transplantation, targeted therapies) are insufficient and what unmet clinical needs this vaccine strategy aims to address. Further, while the authors correctly state that 20–50% of patients relapse, they should clarify whether this statistic applies globally or is specific to a certain region (e.g., the United States).”

The reviewer is right, this information was lacking from our original manuscript. Our revised manuscript now includes a mention of the current leukemia treatments, as well as the problems associated with toxicities and frequent relapses (lines 39-41). Also, the 20-50% relapse rate is a statistic from the United States, which is now specified in the text (line 39).

  1. “Mechanistic Insights and Immune Correlates: While the study effectively demonstrates that ImP is not required for ICV efficacy, the manuscript lacks direct assessment of adaptive immune responses, particularly T cell activation or infiltration. Profiling CD8+ T cells, evaluating tumor-infiltrating lymphocytes, or measuring antigen-specific responses would strengthen the mechanistic link to MHC-I presentation and bolster conclusions on how these vaccines drive anti-tumor immunity.”

We agree with the reviewer, this is a very important point. However, several limitations associated with the L1210 model did not allow for the immune analyses suggested to be performed. For instance, L1210 cells do not form tumor masses, therefore preventing the analysis of immune cell infiltration. Also, the immunogenic epitopes against which anti-L1210 immune responses are directed remain to be identified. Given these limitations, we cannot perform the experiments suggested.

  1. “Clarification of wtVSV-ICV Efficacy: The decreased efficacy of wtVSV-ICV is noted, with speculation on lower inflammation or impaired antigen presentation due to the M protein. However, the manuscript does not directly test these hypotheses. Future experiments investigating cytokine profiles, inflammatory gene signatures, or T cell responses in this context would help clarify this mechanistic gap.”

We also believe that these experiments will be important to fully understand the factors that impair ICV efficacy when using the wt virus. The manuscript includes IFN a and b ELISAs showing that wtVSV, as opposed to oVSV, does not trigger type I IFN production. Future experiments will investigate this aspect is more details.

  1. “Figure Analysis and Quantification: Some western blots (e.g., Figures 2B and 2C) are interpreted as indicating partial induction of PSMB10, yet these claims would be more robust if supported by densitometric quantification normalized to loading controls and presented with statistical analysis. Persistent PSMB10 expression despite IFNAR1 blockade suggests alternative regulatory mechanisms, meriting further investigation (e.g., mRNA analysis, protein stability assays).”

We have now performed densitometric quantifications of all Western blots included in our study. We opted for a normalization over PSMA4 instead of protein loading controls because PSMA4 is found in all proteasomes and therefore allows for a normalization of ImP subunits over total proteasome. Notably, PSMA4 expression did not vary across conditions. Relevant densitometric quantifications can now be found in Figs S1D-E (from Fig. 1F and G), S6A (from Fig. 4A) and S6D (from Fig. 4D). We did not perform statistical analyses because each experiment only shows 1 replicate of each condition. Nevertheless, experiments were performed at least twice (as specified in the corresponding figure legends), and similar findings were obtained.

Also, our results indeed suggest that PSMB10 is differentially regulated compared to PSMB8 and 9. This is already known in the literature with the PSMB8 and 9 genes being located close together and co-regulated, while PSMB10 is encoded on a different chromosome and, while often induced by the same stimuli, it is not always co-expressed with the 2 other ImP subunits.

  1. “Formatting and Presentation: The overall figure quality is low in print; sharper diagrams with clearer labeling are recommended to improve readability. Additionally, minor typographical corrections (e.g., "stain" vs. "strain," "O" in "complete protease inhibitors") should be addressed.”

We noticed that the figure quality is not consistent from one computer to another and depends on the Word software version. High quality images are provided with this resubmission. 

We would like to thank the reviewer for noticing the mistake in section 2.3 (line 126), where we initially wrote Indiana stain. We have now corrected the text to Indiana strain. As for cOmplete protease inhibitors, the company writes the product name with an “O” and we therefore left it as is in the text (line 143).

  1. “Minor Points: There is a formatting issue in section 3.2 (visible gap in paragraph start), which should be corrected for clarity.”

We would like to thank the reviewer noticing this. We have now removed the gap in the first paragraph of section 3.2.

  1. “Line 291–293 suggests a "potential role for ImP expression" in efficacy, which appears inconsistent with the conclusion that ImP is dispensable. This statement should be revised or clarified.”

The reviewer is referring to the following sentence: “Overall, our data using both the L1210 and EL4 models are in line with a potential role for ImP expression for ICV efficacy”. We agree with the reviewer that the sentence was misleading and have now modified it to better reflect the conclusions: “Overall, our data using both the L1210 and EL4 models show that wtVSV is not as effective as oVSV in the ICV setting, which could result from the lack of ImP expression after infection or other differences in the cellular responses to both viruses” (lines 351-352).

  1. “The authors might wish to explicitly discuss the limited generalizability of these findings to human leukemia or to cancers with higher mutational burdens, as acknowledged in the discussion.”

We would like to thank the reviewer for these suggestions. The revised version of our manuscript now includes a mention of the translatability of our findings to human leukemia in the introduction (lines 59-64), as well as an expanded discussion about the potential importance of the ImP for cancers with higher mutational burdens (lines 482-486).

  1. “Conclusions: This study contributes valuable negative data to the field, challenging assumptions regarding the necessity of the immunoproteasome in infected cell vaccine efficacy. It reinforces that while ImP expression is induced and theoretically advantageous for antigen processing, it is not essential in the leukemia models tested. Further mechanistic work examining adaptive immune correlates and extending to more heterogeneous or highly mutated cancers would provide critical insights and improve translatability to clinical settings.”

We would like to thank the reviewer for providing insightful comments and suggestions that improved our manuscript. Future work will investigate highly mutated cancers, as well as alternative mechanisms that could govern ICV efficacy.

Reviewer 2 Report

Comments and Suggestions for Authors

This manuscript presents a well-conducted and scientifically rigorous investigation into the role of the immunoproteasome in the efficacy of leukemia cell-based oncolytic vaccines. The authors make a compelling case that, although immunoproteasome expression is induced by type I interferons following infection with an attenuated vesicular stomatitis virus (oVSV), it is ultimately dispensable for anti-tumor efficacy in murine models of leukemia. This is demonstrated through a thoughtful combination of in vitro mechanistic studies, CRISPR-Cas9 knockout experiments, and in vivo vaccination models.

  1. Line 33: “...is not required for the efficacy...” should be revised to “was not required...” to match past tense used throughout the Results.
  2. The abbreviation "ImP" should be fully spelled out at its first occurrence in the abstract and again in the main text. Similar issues occur with “ICV”, “oVSV”, and “IFN”.
  3. Line 147: The method used for γ-irradiation is briefly mentioned.Provide a rationale for the chosen 30 Gy dose and clarify whether sterility or immunogenicity was evaluated post-irradiation.
  4. In Section 2.2, the concentrations of IFNα, β, and γ are stated. However, no justification or literature references are provided.
  5. Western blot results (e.g., Figure 1B, 1C, Figure 4A) lack densitometric analysis.Add quantitative data normalized to housekeeping proteins to support visual interpretation.
  6. Several supplemental figures are critical (e.g., Figure S1G on ImP subunit expression).Reference these figures more explicitly and incorporate representative data in the main manuscript if critical to claims.

Author Response

  1. “This manuscript presents a well-conducted and scientifically rigorous investigation into the role of the immunoproteasome in the efficacy of leukemia cell-based oncolytic vaccines. The authors make a compelling case that, although immunoproteasome expression is induced by type I interferons following infection with an attenuated vesicular stomatitis virus (oVSV), it is ultimately dispensable for anti-tumor efficacy in murine models of leukemia. This is demonstrated through a thoughtful combination of in vitro mechanistic studies, CRISPR-Cas9 knockout experiments, and in vivo vaccination models.”

We are pleased that the reviewer appreciated the quality of our work.

  1. “Line 33: “...is not required for the efficacy...” should be revised to “was not required...” to match past tense used throughout the Results.”

As suggested by the reviewer, we have now modified the sentence to “was not required” (line 32).

  1. “The abbreviation "ImP" should be fully spelled out at its first occurrence in the abstract and again in the main text. Similar issues occur with “ICV”, “oVSV”, and “IFN”.”

We agree with the reviewer and have now modified the manuscript to define these acronyms in both the abstract and the main text: ImP (lines 23 and 75), ICV (lines 19 and 53), oVSV (lines 25 and 50), IFN (lines 22 and 94).

  1. “Line 147: The method used for γ-irradiation is briefly mentioned. Provide a rationale for the chosen 30 Gy dose and clarify whether sterility or immunogenicity was evaluated post-irradiation.”

This information was indeed lacking from our manuscript. The dose of 30Gy was selected based on the previous study in which the ICV using L1210 cells was first described (reference 5). In their paper, Conrad et. al. show that L1210 cells irradiated following the same protocol lost their proliferative ability and although irradiated cells were not protective on their own, they conferred anti-tumor immunity if infected (as we did in our study: the ICV). Also, another study showed that L1210 cells that were irradiated at doses of 20-40Gy could not proliferate but retained their capacity to secrete cytokines (reference 20). This information is now included in the methods section of our revised manuscript (lines 177-181).

  1. “In Section 2.2, the concentrations of IFNα, β, and γ are stated. However, no justification or literature references are provided.”

The original manuscript indeed lacked justification for the concentrations we used for the different IFNs. The revised manuscript includes references from previous studies by other groups (references 16 and 17).

  1. “Western blot results (e.g., Figure 1B, 1C, Figure 4A) lack densitometric analysis. Add quantitative data normalized to housekeeping proteins to support visual interpretation.”

We have now performed densitometric quantifications of all western blots included in the main figures of our manuscript. We opted to normalize over PSMA4 instead of protein loading controls because PSMA4 is found in all proteasomes and did not vary across conditions. Relevant densitometric quantifications can now be found in Figs S1D-E, S6A and S6D. We did not perform statistical analyses because each experiment only shows 1 replicate for each condition. Notably, experiment were performed at least twice (as specified in the corresponding figure legends) and similar findings were obtained.

  1. “Several supplemental figures are critical (e.g., Figure S1G on ImP subunit expression). Reference these figures more explicitly and incorporate representative data in the main manuscript if critical to claims.”

We agree with the reviewer, some of the data that was initially shown in supplemental figures is critical to the paper and we have therefore moved them to the main figures. As such, Figs S1B, E, F and G now appear in Figure 1 as panels B, C and D, respectively. We would like to thank the reviewer for this suggestion.

Reviewer 3 Report

Comments and Suggestions for Authors

This is an interesting manuscript, reporting on “The immunoproteasome is expressed but dispensable for a leukemia infected cell vaccine” The whole manuscript is informative and accurately reflects the key finding of the study. However, a few recommendations are suggested to further enhance the quality of the report.

Specific comments:

  1. Firstly, consider adding a single line that frames the existing knowledge or hypothesis about immunoproteasome (ImP)’s expected role in infected cell vaccine (ICV) efficacy to better motivate the investigation.
  2. The introduction effectively outlines the burden of leukemia, this extends to a valid rationale that addresses a legitimate knowledge gap in the mechanistic understanding of ICVs.
  3. The methodology is rigorous, well-justified, and appropriate to test the hypothesis, while the relevant methods.
  4. Statistical tests are appropriately used and clearly described in figure legends.
  5. Figures are clear, however addition a table summarizing all in vivo experimental groups, conditions (e.g., prophylactic vs therapeutic), and outcomes would improve reader clarity.
  6. The discussion is clear, however include a short “Future Directions” paragraph to hint at whether the ImP might play a role in antigen processing in tumor types like melanoma or lung cancer where the mutational load is higher.
  7. Limitations that should be acknowledged:
  • C1498 model’s poor immunogenicity may mask subtle effects.
  • The possibility that ImP may be important in high-mutational burden tumors remains to be explored.

Other important revisions that are suggested:

  • Add antibody dilution factors and catalog numbers to western blot methods.
  • Clarify animal handling/randomization and blinding procedures.
  • State number of animals per group and provide power justification, if available.
  • Ensure all graphs have n-values, error bars, and statistical annotations (if possible).
  • Expand discussion on leukemia-specific antigen processing.

Author Response

  1. “This is an interesting manuscript, reporting on “The immunoproteasome is expressed but dispensable for a leukemia infected cell vaccine” The whole manuscript is informative and accurately reflects the key finding of the study. However, a few recommendations are suggested to further enhance the quality of the report.”

We are pleased that the reviewer appreciated the quality of our work.

  1. “Specific comments: Firstly, consider adding a single line that frames the existing knowledge or hypothesis about immunoproteasome (ImP)’s expected role in infected cell vaccine (ICV) efficacy to better motivate the investigation.”

We agree with the reviewer and have now modified the introduction to discuss leukemia tumor antigens that have been shown to be affected by the ImP (lines 83-90 and references 14 and 15). This information strengthens our rationale and we would like to thank the reviewer for this suggestion.

  1. “The introduction effectively outlines the burden of leukemia, this extends to a valid rationale that addresses a legitimate knowledge gap in the mechanistic understanding of ICVs.”

“The methodology is rigorous, well-justified, and appropriate to test the hypothesis, while the relevant methods.”

“Statistical tests are appropriately used and clearly described in figure legends.”

We would like to thank the reviewer for his appreciation of our manuscript.

  1. “Figures are clear, however addition a table summarizing all in vivo experimental groups, conditions (e.g., prophylactic vs therapeutic), and outcomes would improve reader clarity.”

We agree with the reviewer. The revised manuscript now includes a table summarizing the in vivo experimental details, settings and outcomes (Table S1).

  1. “The discussion is clear, however include a short “Future Directions” paragraph to hint at whether the ImP might play a role in antigen processing in tumor types like melanoma or lung cancer where the mutational load is higher.”

The last paragraph of the discussion now includes a discussion about the potential role of the ImP in highly-mutated cancers, as well as future directions on this aspect (lines 481-486).

  1. “Limitations that should be acknowledged:
  • C1498 model’s poor immunogenicity may mask subtle effects.
  • The possibility that ImP may be important in high-mutational burden tumors remains to be explored.”

The revised manuscript now acknowledges these limitations (lines 474-477 and 481-486).

  1. “Other important revisions that are suggested:
  • Add antibody dilution factors and catalog numbers to western blot methods.”

As requested by the reviewer, we have now added all catalogue numbers and dilutions for the Western blot antibodies used in our study (methods section, lines 152-156).

  • “Clarify animal handling/randomization and blinding procedures.”

Animal randomization and blinding was not performed, which is now mentioned in the text (lines 188-189).

  • “State number of animals per group and provide power justification, if available.”

The number of animals per group, as well as exclusion criteria are now stated in the methods section (lines 187-192). The specific number of mice in each group is listed in the figure legends.

  • “Ensure all graphs have n-values, error bars, and statistical annotations (if possible).”

All graphs have n values, error bars and statistical annotations (except for the Western blot quantifications for which only one replicate was quantified).

  • Expand discussion on leukemia-specific antigen processing”

We would like to thank the reviewer for this suggestion. Our revised manuscript now includes information about leukemia-specific antigen processing, which can be found in the introduction section rather than the discussion because we found that it further strengthens our rationale (lines 83-90).

Round 2

Reviewer 1 Report

Comments and Suggestions for Authors

The manuscript is appropriate for publication.